# Exposure to Childhood Healthcare Discrimination and Healthcare Avoidance among Transgender and Gender Independent Adults during a Global Pandemic

**DOI:** 10.3390/ijerph19127440

**Published:** 2022-06-17

**Authors:** Kyle Liam Mason, Shelby A. Smout, Catherine S. J. Wall, B. Ethan Coston, Paul B. Perrin, Eric G. Benotsch

**Affiliations:** 1Health Psychology, Department of Psychology, Virginia Commonwealth University, Richmond, VA 23284, USA; smoutsa@vcu.edu (S.A.S.); wallcs@vcu.edu (C.S.J.W.); pperrin@vcu.edu (P.B.P.); ebenotsch@vcu.edu (E.G.B.); 2Department of Gender, Sexuality, & Women’s Studies, Virginia Commonwealth University, Richmond, VA 23284, USA; bmcoston@vcu.edu

**Keywords:** health, healthcare, healthcare access, healthcare utilization, healthcare discrimination, gender identity, transgender, vulnerable populations, COVID-19 pandemic, United States

## Abstract

Transgender and gender-independent individuals (TGI) encounter myriad barriers to accessing affirming healthcare. Healthcare discrimination and erasure exposure among TGI individuals is vital to understanding healthcare accessibility, utilization behaviors, and health disparities in this population. Exposure to gender identity-related healthcare discrimination and erasure in childhood may contribute to TGI adults’ healthcare utilization behaviors. The commonality of childhood exposure to gender identity-related healthcare discrimination and its relationship to healthcare avoidance during the early months of the COVID-19 pandemic among TGI adults were explored. TGI adults aged 18 to 59 (N = 342) in the United States were recruited online during the summer of 2020. Among individuals who reported childhood exposure to gender identity-related healthcare discrimination, 51% reported experiencing two or more distinct forms of discrimination. Hierarchical logistic regression indicated that exposure to healthcare discrimination in childhood significantly increased the odds of healthcare avoidance during the early months of the COVID-19 pandemic, after accounting for demographic factors and self-reported COVID-19 symptoms (odds ratio = 1.30, 95% confidence interval = 1.10, 1.54). These findings suggest that childhood exposure to gender identity-related healthcare discrimination is a prominent barrier to the utilization of healthcare for TGI adults, even during a global pandemic.

## 1. Introduction

Gender identity-related stigmatization, discrimination, and erasure in healthcare settings serve as central barriers to the accessibility and utilization of healthcare for individuals who embody transgender and gender independent (TGI) identities [1,2]. TGI individuals encounter direct and indirect forms of discrimination in healthcare settings, including healthcare providers’ lack of knowledge about their gender identities and healthcare needs, verbal and physical harassment, abuse, and denial of care [1,3]. Fear-based avoidance of healthcare among TGI individuals reporting exposure to past year healthcare discrimination has been documented [4]. Despite this nascent evidence, the associations between healthcare avoidance in adulthood and childhood exposure to healthcare discrimination among the TGI population are unexplored.

***Terminology.*** The term transgender and gender nonconforming (TGNC) is umbrella terminology used to refer to individuals whose gender identities, gender roles, and gender expressions do not align with or conform to the gender norms associated with their sex assigned at birth [5]. This terminology is commonly considered to be inclusive of the identities that exist within this population, though not all individuals within this population use this terminology to describe their gender identity. In an effort not to reinforce gender binarism (i.e., the idea that gender identity must solely and distinctly conform to assigned sex; [6]) and to affirm the wholeness and internal knowing of gender embodied in individuals within this population; this population is referred to as transgender and gender independent (TGI) in this paper. The term gender independent has been used in previous literature to be inclusive of individuals in this population with non-binary gender identities [7].

***Discrimination, Erasure, Minority Stress, and Healthcare.*** Researchers have utilized the Minority Stress Model to conceptualize the heightened levels of stress, adverse health outcomes, and inefficacious coping strategies (e.g., healthcare avoidance) associated with the chronic, complex, and high levels of gender identity-related discrimination encountered by TGI individuals [8,9]. Exposure to minority stress is related to myriad mental health concerns, including depression, anxiety, and suicidality [2]. Erasure is an additional and distinct form of minority stress that contributes to understanding the intricate systematic, structural, and institutional exclusion that makes this population vulnerable in healthcare settings [10]. Erasure’s impact on this population has been conceptualized in two distinct ways: (1) informational (i.e., the deficit in evidence disseminated regarding the experiences of TGI individuals and the assumption that such evidence is nonexistent), and (2) institutional erasure (e.g., scarcity of policy and structure in organizations and systems to protect or affirm TGI individuals; [10]). Informational erasure can be conceptualized in terms of gender identity data not being routinely collected in population and health-related surveys (e.g., U.S. Census), and institutional erasure can be deciphered when considering medical intake forms that utilize binary sex categories. Although qualitative evidence of their susceptibility to direct and indirect forms of informational and institutional erasure has begun to emerge, knowledge is scant regarding exposure to healthcare erasure in childhood quantitatively among TGI individuals.

Prior work has documented associations between exposure to discrimination and adverse outcomes, including psychological exhaustion, concern for physical and psychological safety, the anticipation of future discriminatory experiences, and avoidance of spaces that may result in discrimination and victimization [3,11]. The prevalence of individual forms of healthcare discrimination has also been documented (e.g., refusal or denial of trans-related care) using dichotomous variable responses (e.g., yes or no [1,3,12]). To the authors’ knowledge, no studies have evaluated the association between childhood exposure to healthcare discrimination and healthcare decision-making behaviors in adulthood among this population. Moreover, there have been no known investigations into exposure to gender identity-related healthcare discrimination in childhood and healthcare utilization behaviors in the early months of the COVID-19 pandemic among this population.

***The COVID-19 Pandemic.*** On 11 March 2020, the World Health Organization (WHO) declared the COVID-19 outbreak a global pandemic [13]. In response, U.S. federal and state governments began to close or restrict access to public entities and issue guidance to the public to mitigate the spread of COVID-19. Beginning in the early months of the COVID-19 pandemic, U.S. healthcare systems have been strained by surges in COVID-19 cases, contributing to a reduction in healthcare resources, facility capacity, personnel, and protective equipment [14]. The COVID-19 pandemic and its strain on U.S. healthcare systems have also been found to contribute to COVID-19-related healthcare avoidance and delay in adults [15]. The U.S. Centers for Disease Control and Prevention (CDC) estimated that by 30 June 2020, 41% of adults in the U.S. had delayed or avoided healthcare due to concerns related to COVID-19 [15]. Researchers have begun to highlight potential individual, structural, and social challenges faced by the TGI population in the context of the COVID-19 pandemic [16,17,18]. Despite the evidence of the impact of the COVID-19 pandemic on the healthcare utilization decisions of U.S. adults generally and the emergence of evidence of the impact of the COVID-19 pandemic on TGI individuals, the association between childhood exposure to healthcare discrimination and healthcare avoidance in adulthood in the context of the COVID-19 pandemic has not been explored.

### Present Study

The present study examined childhood exposure to healthcare discrimination and its relationship to healthcare avoidance due to anticipated gender identity-related discrimination during the early months of the COVID-19 pandemic among TGI adults. It was hypothesized that childhood exposure to healthcare discrimination would significantly predict avoidance of healthcare during the early months of the COVID-19 pandemic due to anticipated gender identity-related discrimination after accounting for other factors that may contribute to healthcare avoidance.

## 2. Materials and Methods

### 2.1. Procedure

Data were derived from participant responses to a survey administered online from 25 June to 4 July 2020. Respondents were recruited using Prolific, an online recruitment platform that connects social, economic, and political science researchers with their intended research demographic. Researchers have found that Prolific offers higher data quality and higher levels of participant naivety and diversity when compared with other online research recruitment platforms [19,20]. Several steps were taken to ensure data quality: (a) respondents completed prescreened demographic information via their Prolific profile to be considered for the study; specifically, the information that respondents reported regarding their assigned natal sex and gender identity when creating their initial Prolific research participant profile was used to invite only those whose gender identities did not align with their assigned natal sex (i.e., not cisgender), (b) IP addresses were examined to ensure that respondents were in the U.S. and to identify potential duplicate responses, (c) the survey platform included survey protection options that allowed the recording of respondents’ anonymous Prolific IDs to identify potential duplicate responses, (d) respondents were required to complete a CAPTCHA challenge to inhibit programmed responses, and (e) respondents were required to answer five attention check questions.

Respondents were eligible if they: (a) were 18 years old or older, (b) identified as TGI, (c) had the ability to complete the anonymous self-administered online Prolific survey in English, and (d) had an approval rating of 95 percent or above in prior research studies completed through the Prolific platform. After participants complete studies on Prolific for which they are eligible, researchers are required to review participants’ data and determine based on specific criteria (e.g., percentage of completion of survey questions, completion of critical survey questions, accurate completion of attention checks) if participants’ submissions should be approved or rejected. The participant approval rating is based on the percentage of approved studies a participant has completed on the platform (i.e., the number of approved studies by the total number of studies an individual has participated in on the platform) [20]. Respondents were prompted to review a consent document prior to accessing the survey. Once consent was obtained and the CAPTCHA question was accurately answered, respondents were asked to answer questions assessing their demographic information, experiences of gender-related discrimination in healthcare, healthcare utilization behaviors, and COVID-19 health impacts. Upon completion of the survey, respondents received compensation for their time in the amount of $1.20. All study materials and procedures were approved by the relevant institutional review board. 

### 2.2. Study Sample

In total, 368 individuals accessed the online survey. Twenty-six participants (7%) were disqualified from the study due to inaccurately responding to the CAPTCHA challenge, not completing any questions beyond the CAPTCHA challenge, inaccurately answering more than one of the five attention check questions, exiting the survey prior to gender identity and assigned natal sex questions, or reporting an apparent cisgender identity. This resulted in a final sample of 342 respondents (93%) being retained for data analyses. Respondents were from 42 U.S. states and the District of Columbia. The U.S. population for 2019 from the U.S. Census Bureau was found to be strongly correlated with the number of respondents from each state, r(49) = 0.95, *p* < 0.001 [21]. On average, respondents were 25.8 years old. The most common gender identities reported among the sample were “non-binary” and “trans man.” Most respondents were Non-Hispanic White had health insurance and reported low annual income. Over half of the sample identified as disabled and neurodivergent. A complete description of demographic information is presented in Table 1.

### 2.3. Measures

*Demographics.* Demographic data were collected regarding age, state of residence, assigned sex, gender identity, annual income, health insurance coverage, disability/neurodivergent identity, and race/ethnicity. A mutually exclusive (Non-Hispanic White or Black, Indigenous, and People of Color [BIPOC] participants) race/ethnicity variable was created for data analysis purposes, consistent with prior research [4].

*Healthcare Discrimination.* Six items were adapted from the healthcare experiences portion of the 2015 U.S. Transgender Survey to assess experiences of healthcare discrimination [1]. Two additional items were added to assess erasure in healthcare settings. Item information is presented in Table 2. Response choices were adopted from the Gender Minority Stress and Resilience Measure (GMSR) and included: “Never”, “Yes, Before 18”, “Yes, After 18”, and “Yes, In the Past Year” [9]. Respondents could choose multiple response choices if they had experienced a specific type of healthcare discrimination at multiple points throughout their lives. A summary variable was created for healthcare discrimination that occurred during an individual’s childhood: the number of events a participant endorsed were totaled, yielding a score with a possible range of 0–8. This measure had adequate internal consistency (a = 0.82) in the present sample.

*Healthcare Avoidance.* An item was adapted from the 2015 U.S. Transgender Survey to assess healthcare avoidance behaviors [1]. The healthcare avoidance item (*“Was there a time since the start of the coronavirus pandemic (11 March 2020) when you needed to see a doctor but did not because you thought you would be disrespected or mistreated as a trans/gender-non-conforming person?”*) assessed respondents’ healthcare avoidance behaviors since the start of the COVID-19 pandemic due to anticipated discrimination. Respondents could select “yes” or “no” to indicate avoidance of healthcare during the early months of the COVID-19 pandemic due to anticipated gender identity-related discrimination.

*COVID-19 Health-Related Questions.* Two yes/no items were adapted from Wang et al. [22] to assess whether respondents had been diagnosed with COVID-19 by a healthcare provider or if respondents had experienced COVID-19 symptoms and had not been tested.

### 2.4. Statistical Analyses

Less than 1% of the data were missing. Preliminary analyses and assumption checks indicated that all normality, univariate and multivariate outlier, linearity, homoscedasticity, and multicollinearity assumptions were met. Data analyses included descriptive statistics and a multivariate analysis. Given the potential influence of many sociodemographic and health status-related factors on healthcare avoidance behaviors [23], six variables (age, race, income, health insurance coverage, disability/neurodivergence, and a variable indicating whether respondents reported experiencing COVID-19 symptoms) were used as covariates in the multivariate analysis. All analyses were conducted using SPSS, version 28.

## 3. Results

### 3.1. Healthcare Discrimination

Thirty-nine percent (n = 134) of the sample reported exposure to gender identity-related discrimination in a healthcare setting during their childhood. Of those who reported childhood exposure to healthcare discrimination, a majority (51%) reported exposure to two or more distinct forms of healthcare discrimination. Detailed childhood healthcare discrimination exposure results are presented in Table 2.

### 3.2. Healthcare Avoidance

Sixteen percent of the sample reported avoiding needed healthcare during the early months of the COVID-19 pandemic due to anticipated discrimination in healthcare settings because of their TGI identity.

### 3.3. COVID-19 Health-Related Questions

Three (0.9%) respondents reported having been told by a healthcare provider that they had COVID-19. Of the 339 respondents who reported that they had not been diagnosed with COVID-19, one in four (25%, n = 89) reported that they had experienced symptoms that might have been COVID-19 (e.g., fever, cough, sore throat, difficulty breathing, or loss of smell), but had not been tested.

### 3.4. Healthcare Discrimination and Avoidance Regression Analysis

A hierarchical logistic regression analysis (Table 3) was conducted to assess whether childhood healthcare discrimination significantly predicted healthcare avoidance in the early months of the pandemic due to anticipated discrimination when controlling for six covariates. The six demographic covariates were entered into the first step of the model, followed by childhood healthcare discrimination in the final step of the model. When the covariates were entered into the model and tested against the constant-only model, they significantly predicted healthcare avoidance in the early months of the COVID-19 pandemic, χ^2^ (6) = 18.01, *p* = 0.01 Nagelkerke R2 = 0.09. When childhood healthcare discrimination was entered into the model, a test of the full model against the constant-only model significantly improved the prediction of healthcare avoidance in the early months of the COVID-19 pandemic, χ^2^ (7) = 26.98, *p* < 0.001, Nagelkerke R2 = 0.13, indicating that together, the covariates and childhood healthcare discrimination reliably distinguished between those who had or had not avoided needed healthcare during the early months of the COVID-19 pandemic, accounting for 13% of the variance. Childhood healthcare discrimination, χ^2^ (1) = 9.56, *p* = 0.002, OR = 1.30, 95% CI [1.10–1.54], significantly predicted healthcare avoidance during the early months of the COVID-19 pandemic over and above the covariates. For each one-point increase in the childhood healthcare discrimination measure, respondents were 30% more likely to have avoided needed healthcare in the early months of the COVID-19 pandemic due to anticipation of gender identity-related discrimination.

## 4. Discussion

This study expands the literature about TGI individuals’ healthcare experiences and utilization behaviors [1,2,3,4,10,12,18]; by documenting high rates of discrimination directed at TGI individuals in childhood, ranging from abusive (e.g., harsh language and refusal of care) to non-inclusive (e.g., lack of identity representation on medical documentation and forms). These findings indicate that TGI youth’s exposure to healthcare discrimination might diminish trust in healthcare institutions and providers necessary for their engagement in care (e.g., preventative care), trust in provider recommendations (e.g., vaccinations), and the improvement of their health outcomes (e.g., disease management) in adulthood. Furthermore, these findings provide evidence of TGI individuals’ experiences in healthcare settings that have the potential to be profoundly harmful.

Independent of an individual’s age, race/ethnicity, income level, disability/neurodivergent identity, and health insurance coverage status, lifetime exposure to healthcare discrimination predicted avoiding needed healthcare in the past year when they anticipated encountering gender identity-based discrimination, even despite this past year including pandemic conditions. Findings indicate that higher levels of lifetime exposure to varied forms of healthcare discrimination significantly increase the likelihood that TGI individuals will avoid needed healthcare due to anticipation of gender identity-based discrimination in healthcare settings. These findings demonstrate the considerable toll that healthcare discrimination has on TGI individuals in instances when they must consider the potential risks of seeking or avoiding healthcare beyond gender-affirming care (e.g., preventative, chronic disease management, sexual and reproductive health, etc.) when it is needed. These findings demonstrate the considerable toll that healthcare discrimination has on TGI individuals in instances when they must consider the potential risks of seeking or avoiding healthcare beyond gender-affirming care (e.g., preventative, chronic disease management, sexual and reproductive health, COVID-19 testing) when it is needed. The postponement and avoidance of healthcare have been found to have detrimental outcomes [15,24]. As one concrete example, previous research has illuminated the associations between delaying needed healthcare and the impact of delayed cancer diagnosis, worsened cancer prognosis, and early mortality in individuals with cancer [24]. What is more, although healthcare avoidance is predominately viewed as a behavior that involves risk at the individual level, it is also important to note that avoiding needed healthcare also presents a public health concern during a global pandemic (e.g., avoiding or delaying COVID-19 testing and vaccinations, contributing to higher transmissibility as well as the personal risk of hospitalization and death) [14,15].

### 4.1. Implications for Clinical Practice

Healthcare institutions and providers can work to ensure that these barriers of cisgenderism (i.e., the delegitimization of individuals’ internal knowing of their genders and bodies) and cisnormativity (i.e., the assumption that all individuals are cisgender) are replaced with affirmation and a conceptualization of gender as a more expansive, nonbinary construct [25]. It is imperative that healthcare providers attend to their individual appraisals and biases about gender identity and expression, as such biases and appraisals may impact the quality of care provided to TGI individuals. Healthcare institutions should ensure that the entirety of healthcare teams (e.g., intake staff, billing staff, and providers) are provided with the training necessary to provide quality care for this population. TGI individuals should not be burdened with providing education to their healthcare providers to have their healthcare needs met.

Inclusive forms and records would also support TGI individuals in conveying the manner in which their gender identity intersects with their bodies to healthcare providers. TGI individuals may conceal their gender identities to avoid discrimination in healthcare settings which has healthcare accessibility (e.g., gender-affirming care) and outcome (e.g., receiving inappropriate care) implications [10]. The use of inclusive intake forms may serve as an affirming signal to TGI individuals that their embodied identity is valued by healthcare providers (i.e., identity safety cues; [26]). Health providers should attend to the preferred names, gender identity terms, and pronouns that TGI individuals regard most representative of their identities.

### 4.2. Implications for Policy

TGI individuals, particularly youth, their families, and their doctors, are being increasingly targeted by legislation and executive orders issued by governors that restrict the accessibility of gender-affirming care in states across the U.S. [27]. Despite these restrictions being opposed by nearly all reputable medical organizations (e.g., American Psychological Association, American Academy of Child and Adolescent Psychiatry, American Academy of Pediatrics, Endocrine Society, American College of Obstetricians and Gynecologists, and World Professional Association for Transgender Health), 17 states have 23 active bills specifically targeting TGI youth healthcare, their consenting parents, and their providers [27]. Legal and policy frameworks ensuring the protection of TGI individuals are vital to ameliorating the harm these findings have articulated. Though recent advancements have been made, such as the U.S. Supreme Court ruling that employers cannot terminate TGI individuals based on their gender identity [28], policies that make explicit the illegality of healthcare discrimination in all its forms must be enacted.

### 4.3. Limitations and Implication for Future Research

Causal inferences could not be made in this study due to its cross-sectional design. Longitudinal cohort studies should be employed to expand knowledge regarding TGI identity development and healthcare accessibility barriers. This study used an online and non-probability sampling method, thereby lessening its representativeness of the diversity that exists among the TGI population. Despite the geographical representativeness of the U.S, the most marginalized tier (e.g., BIPOC, Limited English Proficiency, and economically vulnerable individuals) of this population were unable to be accessed; as such, these results cannot be generalized to all TGI individuals. Future studies should strive to utilize probability sampling. While this study captured the reported prevalence of healthcare discrimination and healthcare avoidance behaviors associated with the discrimination faced in healthcare settings by this population, it did not investigate exposure to discrimination in specific healthcare settings (e.g., gynecology, oncology, endocrinology, psychology). Future studies should assess exposure to erasure and discrimination within specific healthcare specialties. This study garnered information about respondents’ disabilities/neurodivergence but did not inquire about participants’ diagnoses of specific chronic health conditions.

## 5. Conclusions

Notwithstanding these limitations, this study provides information that is critical to understanding the health behaviors and subsequent health implications for the TGI population. This study provides evidence of the persistence of myriad institutional, structural, and ideological barriers to transgender and gender-independent individuals’ abilities to access healthcare, with associations of deleterious health behaviors (e.g., avoidance) generally, amidst a global pandemic, and has articulated numerous clinical and policy implications for future consideration.

## Figures and Tables

**Table 1 ijerph-19-07440-t001:** Sample Demographics.

Variable	n (%)
Age: min-max, mean (SD). 18–59,	25.8 (7.2)
Assigned Natal Sex	
Female	246 (72.1)
Male	95 (27.9)
Gender Identity	
Non-Binary	134 (39.2)
Trans Man/Man	42 (12.3)
Trans Woman/Woman	33 (9.6)
Another Identity Not listed	32 (9.4)
Man	23 (6.7)
Woman	23 (6.7)
Gender Fluid	22 (6.4)
Genderqueer	21 (6.1)
Gender non-conforming	12 (3.5)
Race or Ethnicity	
Non-Hispanic White	235 (68.7)
Multiracial/ethnic	48 (14.0)
Asian/Asian American	21 (6.1)
Hispanic/Latinx	20 (5.8)
Black	15 (4.4)
Native American	2 (0.6)
Other race/ethnicity	1 (0.3)
Annual Income	
≤$20,000	114 (33.3)
$20,001–$40,000	81 (23.7)
$40,001–$60,000	44 (12.9)
$60,001–$80,000	35 (10.2)
$80,001–$100,000	22 (6.4)
$100,000 <	46 (13.5)
Health Insurance Coverage	
Yes	276 (80.7)
No	66 (19.3)
Disability/Neurodivergent Identity	
Yes	180 (52.6)
No	162 (47.4)

N = 342.

**Table 2 ijerph-19-07440-t002:** Childhood Healthcare Discrimination Exposure.

*Item*	n (%)
*I had to teach a doctor or other health care provider about trans/gender non-conforming people so that I could get appropriate care.*	48 (35.8)
*A doctor or other health care provider refused to give me trans/gender non-conforming-related care.*	31 (23.1)
*A doctor or other health care provider refused to give me other health care (such as a physical exam, flu, diabetes).*	11 (8.2)
*A doctor asked me unnecessary/invasive questions about my trans/gender non-conforming status that were not related to the reason for my visit.*	34 (25.4)
*A doctor or other health care provider used harsh or abusive language when treating me.*	28 (20.9)
*I was verbally harassed in a health care setting (such as a hospital, office, clinic).*	23 (17.2)
*A doctor or other health care provider refused to use the pronouns or names that I requested to be used.*	47 (35.1)
*The medical forms or documents that a doctor or other health care provider asked me to complete did not include my gender identity.*	108 (80.6)

N = 134.

**Table 3 ijerph-19-07440-t003:** Logistic Regression Analysis: Childhood exposure to healthcare discrimination and healthcare avoidance during the early months of the COVID-19 pandemic.

Post Start of COVID-19 Pandemic Healthcare Avoidance AOR, (95%, CI)	
Healthcare Discrimination	1.30 *, (1.10–1.54)

AOR: Adjusted Odds Ratios (values reported are adjusted for the influence of age, race, income, health insurance, disability, and self-reported COVID-19 symptoms); CI = confidence interval. N = 342, * *p* < 0.05.

## Data Availability

The data used and analyzed in this study are available from the corresponding author on reasonable request. The data are not publicly available due to privacy and confidentiality concerns and the sensitive nature of the data collected from the sample.

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
