# Peer review of "Exposure to Childhood Healthcare Discrimination and Healthcare Avoidance among Transgender and Gender Independent Adults during a Global Pandemic"

_ijerph, 2022, doi:10.3390/ijerph19127440_

Round 1

Reviewer 1 Report

This is a highly interesting and innovative manuscript. Below are my suggestions and comments

I have just a few questions for the authors:

  1. Title is too long, please summarize it
  2. Table 1 (SES) should include participants age (mean & SD)
  3. Is it possible to describe the prevalence of healthcare avoidance among those with a positive COVID-diagnosis?
  4. Healthcare avoidance can be influenced by a broad range of factors beyond childhood discrimination. For instance, those who experienced racism in a healthcare setting (a very common experience for People of Color), or those who were living in deep poverty, foster homes or experiencing house instability during childhood might be more at risk of experiencing discrimination in healthcare. On the same note, those experiencing deep poverty, homelessness, house instability nowadays might be more prone to avoid healthcare facilities as well...
  5. Is there any difference between participants living in more liberal states vs. those who are living in more conservative ones nowadays? What about during childhood, is childhood discrimination in healthcare facilities more frequent in a specific group of states? Is it possible to explore this with your dataset?
  6. The discussion section is too confusing. Here you should compare your findings with other similar studies. There are several studies trying to better understand the roots of healthcare discrimination faced by LGBTQ+ persons, please include a few studies here and compare your findings. 
  7. Discussion, 3rd paragraph: "Given the current sociopolitical climate" - this statement is too vague for a scientific manuscript. Please explain what are you referring here, citing sources that underscore your idea. If you are mentioning the politicization of gender-affirming health care in the US, please cite it. If you are referring to the potential criminalization of healthcare professionals who care for TGI, please state so. International readers might not be aware of the 'current sociopolitical climate' in the US, and even if they are, it is important to mention.
  8. Recall bias should be mention as a limitation of your study
  9. Conclusion: although the discussion here is highly important and timely, it is beyond the scope of your study, making this section quite confusing.  Both conclusion and discussion sections should be reorganized and focus more on your study. One or two final paragraphs could discuss the impact of current repressive legislations being adopted by the US, but this cannot be the main aspect discussed here. The key point seems to be how childhood discrimination experiences in healthcare facilities might influence care avoidance during the COVID-19 pandemic - this is the core of your discussion/conclusion.

Thanks for the opportunity to review it. 

Reviewer 2 Report

This manuscript describes a cross-sectional, survey-based research project with the goal of determining whether experience of childhood gender-identity-related healthcare discrimination impact the use of healthcare services in gender-expansive adults during the early part of the COVID-19 pandemic. The authors developed an online survey distributed anonymously to their target population, with questions specifically designed to assess childhood discrimination experiences and their influence on healthcare utilization in adulthoods in the target population. The results suggest that in this population, experiences of healthcare discrimination during childhood, specifically related to gender-identity do impact the timing and tendency toward use of the healthcare system in adulthood in gender expansive individuals.

- Although it is well documented that transgender and gender expansive individuals have a lower rate of healthcare utilization than cisgender people, this study adds to the literature that there is a correlation between this finding and childhood discrimination experiences.  

- I'm not sure whether the fact that this occurred in the setting of the earlier months of the COVID-19 pandemic is relevant; it seems likely that these individuals would avoid healthcare regardless of the pandemic if the reason is based on previous negative experiences in the healthcare setting.

- It is also known that most humans were less likely to seek healthcare during the early days of the COVID-19 pandemic, so comparing cisgender healthcare utilization to transgender and gender expansive utilization would be been useful to tease this out further.

- Although these results are significant and worthy of publication, it should be noted that the respondents are a select group - one that had access both to the internet in general, and the survey platform specifically. The authors did address this, but knowing that a large portion of the transgender and gender expansive population is not represented in this study is important.

- Overall I do believe this knowledge is worthy of dissemination, and recommend that with some copy-editing addressing several issues with grammar, terms, and punctuation this article should be published.

Reviewer 3 Report

This paper aimed to assess the association between childhood exposure of gender identity-related healthcare discrimination and healthcare avoidance during adulthood. This research topic is highly relevant and therefore I appreciate the opportunity to review this paper. However, the current manuscript has some methodological limitations that should be addressed:

1.      In the eligibility criteria (d), it was unclear how the approval rating was obtained in prior research studies and what a 95% approval means. The authors should explain this in more detail.

2.      The authors mix the terms “healthcare underutilization” with “healthcare avoidance” throughout the manuscript.  “Healthcare underutilization” can be both active and passive- for instance the person does not know about healthcare services and may therefore not used them (rather passive) or the person actively does not use the services, although they may know about them (active). Healthcare avoidance is rather active. I would suggest sticking to one term (healthcare avoidance) throughout for the title, abstract, and text.

3.      Related to above, it was unclear to me how “healthcare avoidance” was measured. The authors cite the 2015 US Transgender Survey as having the items to assess healthcare avoidance behaviors, but did not mention the items were used (nor could I find them). I would ask the authors to list the items as well as how they created the corresponding variable assessing healthcare avoidance, similar to how they described “healthcare discrimination”.

4.      Regarding the multivariate analysis, why did the authors include self-reported COVID-19 symptoms in the analysis? I don’t see how this is a confounder in the relationship between healthcare discrimination and healthcare avoidance. I would suggest omitting it from the analysis.

5.      The detailed text in the conclusions belongs in the discussion. The conclusions should briefly summarize what was learned from the study and possible policy implications.

6.      Given the study limitations (selection bias, cross-sectional design), some statements should be weakened, such as Line 323 “As this study confirms…” etc.

Some minor points:

1.      Line 283: Lessening instead of lessoning

2.      Lines 204-205: incomplete sentence

Round 2

Reviewer 3 Report

Dear authors,

thank you for your revisions- this version looks much improved. However, like I mentioned in my previous review, I would like to see the conclusions much shortened (a paragraph or two short paragraphs), and the policy section also shortened and moved to the discussion in a section "policy implications". This is a scientific paper, and as much as I may agree with you have written, I think that much of the discourse in the conclusions can be significantly shortened and put into the context of an international scientific audience.

Thanks again for the opportunity to review this interesting paper.

Author Response

Thank you for your feedback regarding the discussion and conclusion sections of this manuscript. The discussion and conclusion sections have been overhauled. The clinical and policy implications have been moved to the discussion section – as per your suggestion. Key points have been reorganized without an overwhelming among of U.S. policy information, though some was retained – as the authors feel that this is a vital context to capture. The absence of protective policies – has in many ways allowed for the permissibility of the healthcare discrimination that this sample endorsed. The conclusions have been described in one paragraph as per suggestion. We do hope that the revision of these sections is acceptable to the reviewer, as we took good care to strike both a scientific balance and advocacy for a pathway through which such manuscripts about healthcare discrimination – may not be needed as much in the future – with the support of clinical and policy changes.